# Inhibition of *xpt* Guanine Riboswitch by a synthetic nucleoside analog

Swapan S. Jain[1]*, Emily C. McLaughlin[1]*, Gabriel G. Perron[2,3],
Mallikarjunachari Uppuladinne[4,5], Seoyoung Kim[1], Katherina Gindinova[1],
Silvie H. Lundgren[1], Liad Elmelech[1], Uddhavesh Sonavane[4], Rajendra Joshi[4],
Korrapati Narasimhulu[5]

1 Chemistry and Biochemistry Program, Bard College, New York, United States of America, 2 Center for Genomics and Systems Biology, New York University, New York, United States of America, 3 Biology Program, Bard College, New York, United States of America, 4 High Performance Computing - Medical and Bioinformatics Applications Group, Centre for Development of Advanced Computing (C-DAC), Pune, Maharashtra, India, 5 Department of Biotechnology, National Institute of Technology Warangal, Warangal, Telangana, India

* sjain@bard.edu (SSJ); mclaughl@bard.edu (ECM)

## Abstract

Riboswitches are structured elements predominantly found in the 5'-untranslated region of many bacterial mRNA. These noncoding RNA regions play a vital role in bacterial metabolism and overall function. Each riboswitch binds to a specific small molecule and causes conformational changes in the mRNA leading to regulation of transcription or translation. In this work, we have synthesized SK4, a novel nucleoside analog that binds to the guanine riboswitch mRNA of the xanthine phosphoribosyl transferase gene in *Bacillus subtilis* and terminates transcription of the riboswitch mRNA to a greater extent than the native ligand guanine. Molecular dynamics simulations of SK4 with riboswitch mRNA reveal an overall stable complex with additional bonding interactions in comparison to guanine. Our work with SK4 demonstrates that specific genes in bacteria can be effectively controlled by ligand analogs, providing an alternative mechanism to regulate the function of bacteria.

## Introduction

RNA has the ability to form complex structures and conformations which can be exploited by targeted drug design [1–4]. Riboswitches are regions within the untranslated sequence of certain bacterial mRNA genes that can act as powerful gene regulatory elements [5]. Riboswitches are a class of noncoding RNA that do not require proteins for their function. This finding gave credibility to the RNA World hypothesis which postulates that RNA was a precursor to DNA and proteins having the ability to not only transmit genetic information but also carry out catalytic activity [6]. Riboswitches are highly specific in the small molecule that binds to the sensing domain called the aptamer domain (Fig 1A). This leads to conformational changes in the

**Data availability statement:** All relevant data are within the manuscript and its Supporting Information files. We confirm that the submission contains our minimal data set.

**Funding:** The authors acknowledge financial support from Research Corporation for Science Advancement Award # 21054 to SSJ. Generous funding for this project was also provided by the Office of Undergraduate Research at Bard College, Bard Summer Research Institute, and the Chemistry & Biochemistry Program at Bard College. There was no additional external funding received for this study.

**Competing interests:** The authors have declared that no competing interests exist.

adjacent actuator domain called the expression platform. Structural changes in the expression platform causes regulation of transcription, translation, and other genetic processes [7–9]. More than 50 classes of riboswitches have been discovered in the last two decades [10]. These riboswitches are capable of selectively binding small ions, amino acids, metals, coenzymes, sugars, nucleobases, and nucleotide derivatives [11–17]. Riboswitches are predominantly found in bacteria and it is estimated that more than 2% of genes in certain species of bacteria may be under riboswitch regulation [15]. TPP (thiamine pyrophosphate) riboswitches are the only one of its kind to have also been reliably found in algae, fungi, and plants [16–18].

Due to their near exclusive presence in bacteria, riboswitches represent an attractive opportunity for drug targeting by small molecules. The rise of antibiotic-resistant bacteria such as Methicillin-resistant *Staphylococcus aureus* and *Clostridium difficile* is a public health crisis that will lead to millions of global deaths by 2050 [19–20]. Infectious bacteria are targeted by existing drugs that function by identical or greatly similar mechanisms (binding to proteins or enzymes to disrupt DNA replication, ribosome function, cell wall synthesis); targeting of bacterial mRNA by new types of drugs validates riboswitches as drug targets [21]. However, targeting of riboswitches in bacteria is relatively underexplored. Guanine and adenine riboswitches were the first to be discovered in bacteria [22]. The messenger RNA of the *xpt* gene in bacteria codes for the xanthine phosphoribosyl transferase enzyme which catalyzes the synthesis of xanthine monophosphate from xanthine. Batey and Breaker have both demonstrated that guanine binding to the riboswitch aptamer domain leads to the formation of a distinctly folded RNA segment in the expression domain responsible for transcription termination [23,24].

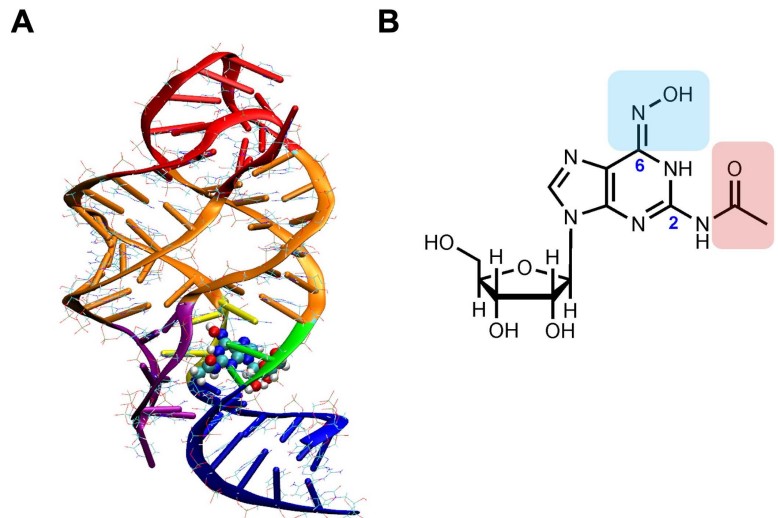

**Fig 1. Structure of the guanine riboswitch aptamer domain and SK4 analog. (A)** Cartoon representation of the aptamer domain of the *xpt* riboswitch in complex with SK4 (colored spheres). Riboswitch regions are color-coded as follows: P1 (blue), P2, P3 (orange), L1, L2 (red), J1-2 (yellow), J2-3 (violet) and J3-1 (green) where P = paired, L = loop, J = junction **(B)** Chemical structure of SK4 showing modifications at the C2 (red) and C6 (blue) positions. Figures were created using VMD software (A) and ChemDraw Professional 23.0 **(B)**.

A study done by Breaker's group modified guanine at either the C2 or the C6 position of the purine ring using a variety of functional groups [25]. They demonstrated that some of the analogs bound to the riboswitch aptamer domain with a greater affinity than guanine. Batey and coworkers have also designed purine analogs with iterative modifications at the C2, C6, and C8 positions that rival the binding affinity of guanine [26]. In 2015, we reported the design and synthesis of two unique guanine analogs with simultaneous modification at both the C2 and C6 positions (C2 N-acetyl/ C6 methyl-amine and C2 N-hydrazone/ C6 hydrazone) affording even more sites for hydrogen bonding [27]. Even though these modifications led to an improvement in the binding affinity of the analogs to the guanine riboswitch; subsequent *in vivo* work showed that the synthesized analogs were less effective at inhibiting the riboswitch expression than the parent compound guanine. We rationalized this seemingly negative result as a consequence of poor aqueous solubility of the synthetic analogs. Lower solubility of our analogs lead to a reduced uptake inside the cells and subsequent bioavailability at the aptamer binding site.

Herein, we report the synthesis of a novel nucleoside analog SK4 with three modifications on the purine ring: (i) a hydroxylamine group (-NH-OH) at the C6 position, (ii) an N-acetyl group (-NH-COCH$_3$) at the C2 position, and (iii) a ribose sugar attached to the N9 position (Fig 1B). Our motivation behind these modifications was not only to improve solubility by adding a ribose sugar but also to modify the purine ring of the analog to endow it with additional bonding capabilities with the nucleotides in the riboswitch aptamer domain. Work by LaFontaine has shown that pyrimidine analogs that present hydrogen bonding groups at similar positions as guanine bound tightly to the riboswitch [28]. A significant body of work done by Breaker's group has shown that variant riboswitches with similar architecture to the *xpt* riboswitch can bind to a variety of unique ligands including deoxyribonucleosides [29–32]. Moreover, tautomeric variations in the ligand functional groups, such as the C6 hydroxylamine in this work, can heavily influence binding to the riboswitch [33]. In this work, we have used *B. subtilis* reporter gene assays and quantitative PCR to show that SK4 lowers expression of the riboswitch mRNA to a greater extent than guanine and guanosine. Molecular dynamics simulations also demonstrate that SK4 binds to riboswitch RNA with strong binding affinity and additional hydrogen bonding interactions with the new functional groups and the ribose sugar. These are promising results that demonstrate the potential of analog binding to riboswitches as a potential mechanism of bacterial gene regulation.

## Results and discussion

Binding of nucleobase analogs to guanine riboswitches has been investigated previously [10]. In this work, we have synthesized SK4 (Fig 1B), a nucleoside analog capable of binding to the guanine riboswitch. The abundance of nitrogen atoms and three hydroxyl groups due to the presence of ribose sugar make SK4 highly polar, more soluble in an aqueous medium, and provides ample sites for hydrogen bonding, compared to our previously studied derivatives. More specifically, we designed this nucleoside analog to feature functional groups at the C2 and C6 position of the purine ring to further increase hydrogen bonding capability with RNA bases in the riboswitch. The synthesis of SK4 was accomplished, starting from commercially available guanosine (Fig 2). From guanosine, a two-step protocol of acetylation of the ribose hydroxyl groups and subsequent chlorination at the C6 position resulted in modest yield of SK2 [34]. The C2 primary amine was subsequently acetylated to yield SK3 and the nucleophilic aromatic substitution of chlorine for hydroxylamine at C6 was relatively facile, affording SK4-OAc in mildly basic conditions. However, deprotection of the ribose hydroxyl groups remained the greatest challenge in this synthesis, as most deacetylation trials led to mixtures of SK4-OAc and SK4 that were difficult to purify. Finally, when SK3 was treated with ethanolic potassium hydroxide and a large excess of hydroxyl amine, the deprotection and substitution were both realized in one reaction flask, providing the novel guanosine analog, SK4. The structure was confirmed and fully characterized by NMR, IR, and mass spectrometry (S1–S5 Figs).

After the successful synthesis of SK4, we carried out *in vivo* experiments using *B. subtilis* cells where the beta galactosidase reporter gene was inserted downstream of the *xpt* riboswitch gene. *B. subtilis* cells containing the *xpt*-galactosidase fusion cassette were graciously provided to us by the Breaker Lab at Yale University. Cells were grown

**Fig 2. Synthesis of SK4.** Analog SK4 was prepared in a four-step protocol starting from commercially available guanosine. The synthesis required acetyl protection (SK1) of the ribose hydroxyl groups and chlorination at C6 (SK2) before the desired modifications were made at C2 and C6 (SK3 and SK4).

in the presence of guanine, guanosine, and SK4 ligands at varying concentrations. Ligand binding caused structural changes in the aptamer domain of the *xpt* riboswitch [24]. These conformational changes led to premature termination of transcription which also causes transcription termination of the downstream galactosidase mRNA. Lower messenger RNA levels resulted in lower levels of galactosidase enzyme which can be quantified using a colorimetric UV-Vis assay [35]. Our results show that SK4 is more effective in lowering mRNA expression in comparison to guanosine and even guanine (Fig 3). At 25 μM and 50 μM, SK4 binding leads to the termination of mRNA transcription to a statistically significant greater extent compared to guanine and guanosine. This finding was surprising because guanine is a native ligand for the *xpt* riboswitch. In our view, an improvement in solubility by adding a ribose sugar moiety contributes to increased bioavailability of the analog at its RNA binding site. Coupled with modifications at C2 and C6 positions of the purine ring, we have a compound with tight binding affinity as well as improved solubility and delivery in a cellular context.

We wanted to further investigate this surprising result where SK4, a synthetic nucleoside analog, was able to lower the expression of the *xpt* riboswitch mRNA to a greater extent than the native ligand guanine. In addition to using a reporter gene assay such as beta galactosidase; direct quantification of the amounts of transcribed riboswitch RNA in the presence of SK4 is invaluable. Therefore, we carried out RT-qPCR experiments where *B. subtilis* cells were grown in the presence of 50 μM guanine, guanosine, and SK4. Subsequently, total RNA was isolated from the cells and RT-qPCR experiments were performed in order to amplify a 120-base coding region of the *xpt* mRNA which is downstream of the riboswitch region. A greater degree of transcription termination should yield lower amounts of RNA transcripts. In a quantitative PCR experiment, lower amounts of starting RNA template would be manifested as a higher threshold Cq (quantitative cycle) value which means that a greater number of cycles would be needed in order to reach the same threshold fluorescence yield (i.e., similar amounts of amplicons) compared to other samples. Our results in Fig 4 demonstrate that *B. subtilis* cells grown in the presence of SK4 yield the lowest amount of starting RNA transcripts (template RNA) which is indicated by

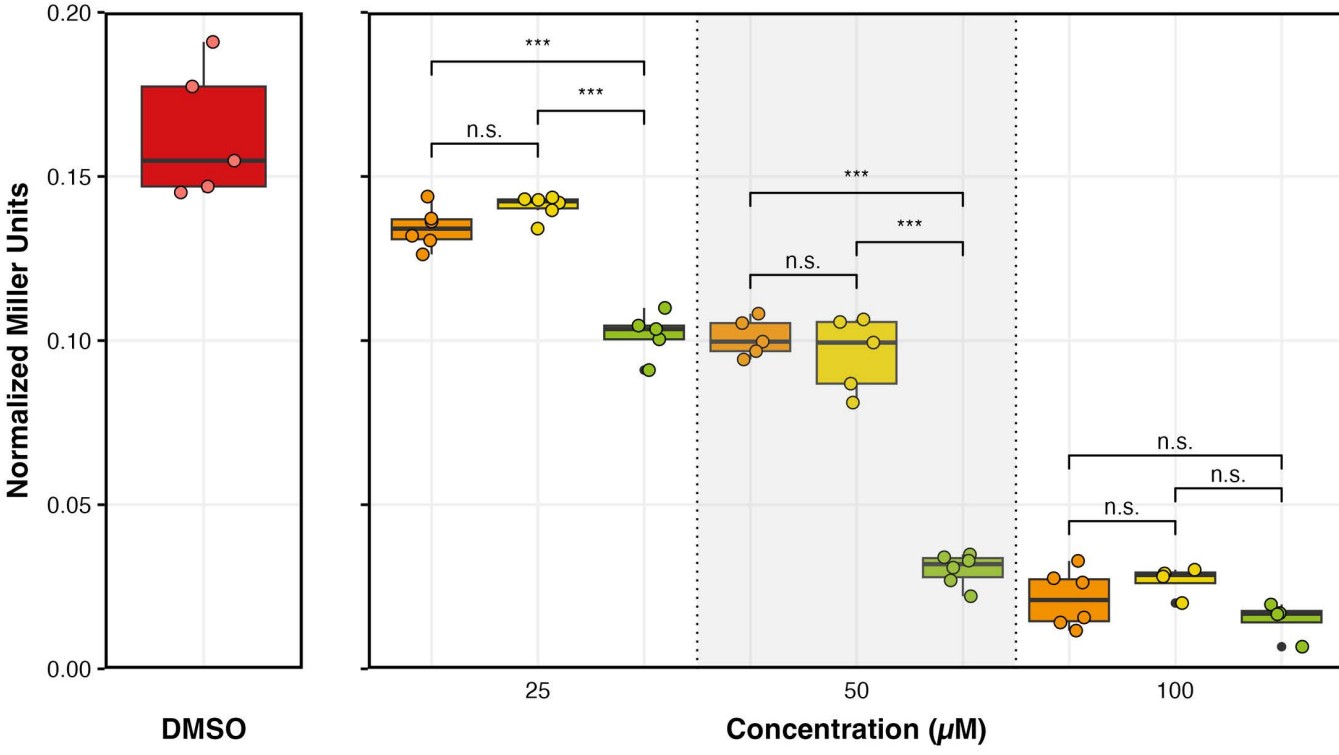

**Fig 3. Miller assay results in the presence of ligand analogs.** Beta galactosidase reporter gene assay showing the expression of guanine riboswitch in the presence of DMSO control (red), guanine (orange), guanosine (yellow), and SK4 (green). Samples were all different from the DMSO control and were tested at three different concentrations. Significance differences were estimated using an analysis of variance followed by a post-hoc Tukey test (i.e., P < 0.0001, ***, P > 0.5, n.s.). Box plots show the median as well as the interquartile range. Individual points represent normalized miller units for each trial.

the largest Cq value (27.3). Cq values for guanine (26.2) and guanosine (26.0) indicate the presence of a larger amount of starting RNA transcripts in comparison to SK4, with a roughly two-fold decrease in expression in the presence of SK4 when compared to the other two compounds. These results are consistent with our reporter gene results in Fig 3 which also show that SK4 binding to the aptamer domain of the riboswitch leads to a greater degree of RNA transcription termination. Although RNA copy numbers have not been measured using a standard curve assay, the statistically significant differences in Cq values (Fig 4) between SK4 - guanine and SK4 - guanosine indicate that SK4 is significantly more effective in lowering transcription of the *xpt* gene.

It is important to note that the viability of *Bacillus subtilis* cells is not impacted by SK4. Kirby Bauer disc diffusion assays demonstrate uninhibited cell growth up to a concentration of 500 μM SK4 (S6 and S7 Figs). Previous work by others has shown that guanine riboswitch variants can evolve to recognize 2'-deoxyguanosine [31,32]. Mutations in L2 and L3 loops which are important in riboswitch folding as well as changes in J1-2 and J2-3 junctions can lead to the adoption of a variant structure enabling recognition of 2'-deoxyguanosine. We do not suggest that the *xpt* mRNA has undergone mutations to afford novel interactions with SK4. Rather, improved solubility, bioavailability, and additional interactions due to functional group modifications at C2 and C6 position likely causes an increase in binding of SK4 with the aptamer domain of *xpt* mRNA and termination of transcription.

We wanted to investigate the potential binding interactions of SK4 with the aptamer domain that could explain its inhibition of the riboswitch RNA. Therefore, molecular docking and molecular dynamics (MD) simulations were performed to understand the binding affinity SK4, guanine, and guanosine to the aptamer domain of the riboswitch and to understand

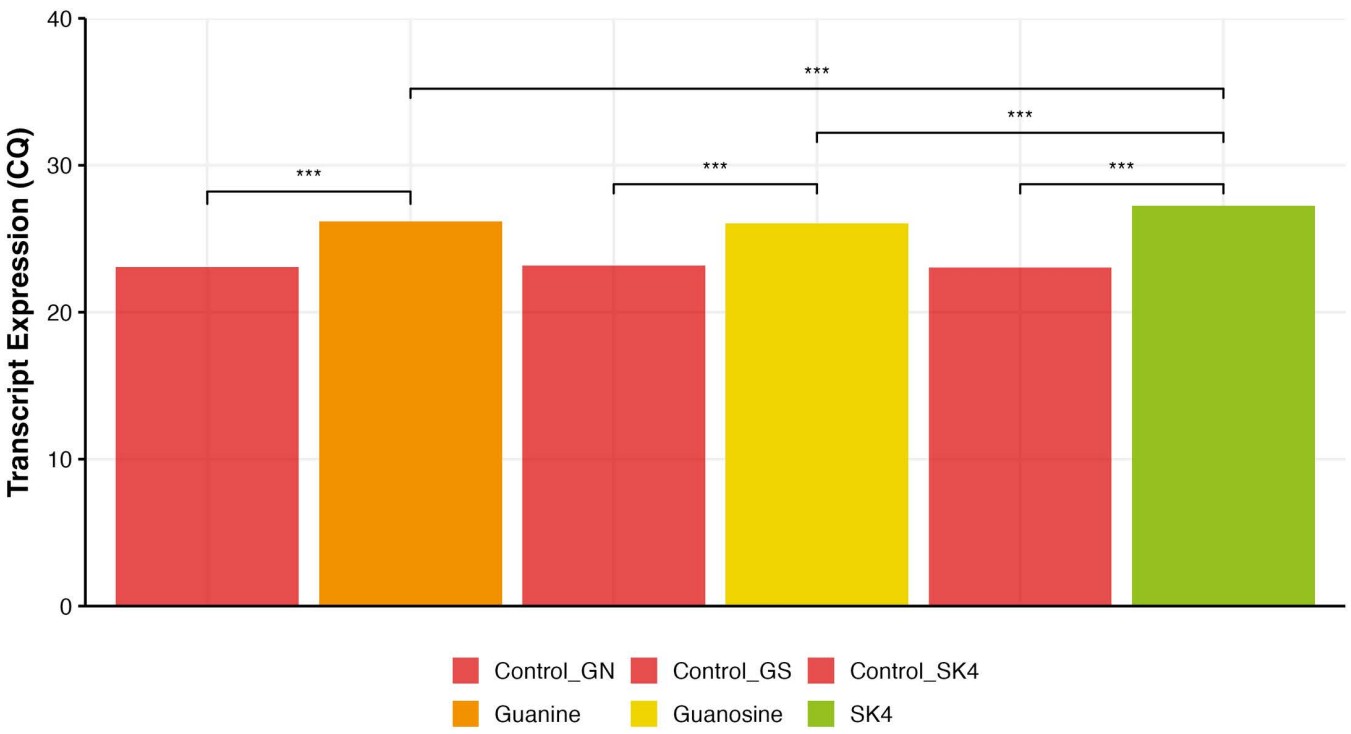

**Fig 4. RT qPCR riboswitch expression in the presence of various ligands.** Quantitative PCR was used to determine expression of riboswitch RNA in *B. subtilis* cells at 50 µM guanine (orange), guanosine (yellow) and SK4 (green). GAPDH mRNA expression was utilized as an internal control for all three samples where GN is guanine and GNS is guanosine. All samples were run in quadruplicates and were different from their respective controls (red). Crucially, SK4 was also statistically different from Guanosine and Guanine. Significance differences were estimated using an analysis of variance followed by a post-hoc Tukey test (i.e., $P < 0.0001$, ***, $P > 0.5$, n.s.).

the dynamic behavior of the riboswitch upon binding to these ligands. The docking scores for guanine, guanosine and SK4 are -28.82, -31.63 and -55.81 kcal/mol, respectively. According to the initial docking scores, SK4 binds strongly to the riboswitch compared to its cognate ligand guanine as well as guanosine. MD simulations were carried out in explicit solvent at 1000 ns timescale. The room mean square deviation (RMSD) values with respect to the starting structure for all the systems were between 2–7 Å (Fig 5A) with an average value range of 2–4 Å. Importantly, the riboswitch bound ligands showed RMSD values below 3 Å. Small RMSD values in a simulation indicate that the system has attained equilibrium. The four systems, NoLig, GUA-docked, GNS-docked, and Ensemble SK4-docked maintained stable RMSD values while other two systems Crystal and SK4-docked show RMSD fluctuations in the middle of the simulations which settle down during the latter stages of the simulation. RMSF (room mean square fluctuations) is the residue-wise fluctuation analysis of the riboswitch structure from the 1st nucleotide to the final nucleotide. RMSF were observed to be high at the loop (L) regions and junction (J1-2, J2-3, J3-1) regions compared to paired (P) regions (Fig 5B). The high RMSF deviations were observed between different systems at junction regions because the major binding pocket of the riboswitch consists of junction regions. The high fluctuations at the junction regions may be due to the rearrangement of the binding pocket to accommodate the bulky ligands guanosine and SK4. Our analysis in Fig 5B shows that the simulation timescale does not cause any significant changes in the region-wise structure of the riboswitch upon SK4 binding in comparison to guanine or guanosine.

Molecular mechanics free energy calculations show that SK4 bound to the riboswitch structure with a lower overall free energy (-10.9 kJ/mol) than guanosine (-5.1 kJ/mol); however the parent compound guanine yielded the most stable structure with the lowest free energy of -28.7 kJ/mol. This is explained by the observation that the number of strongly

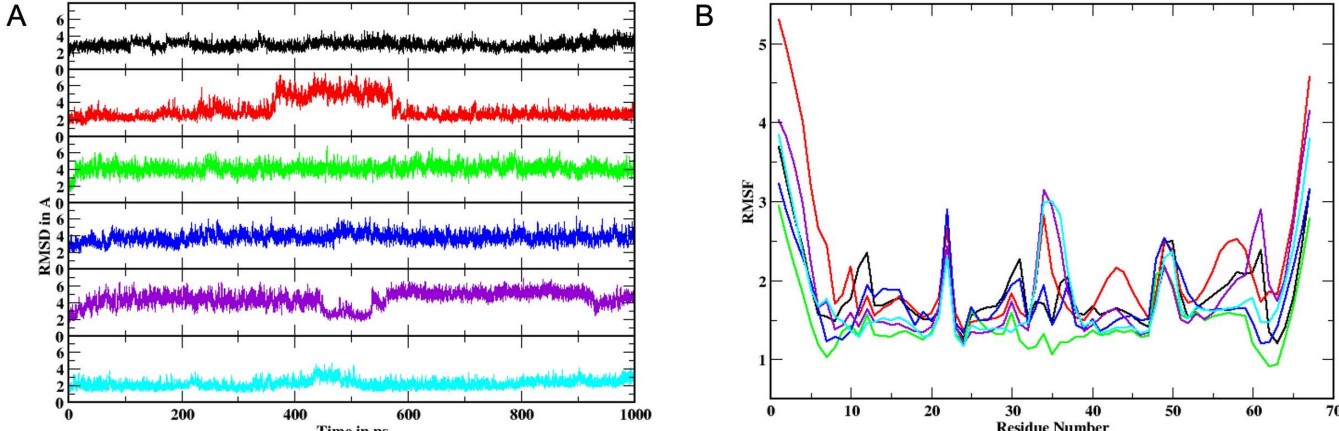

**Fig 5. Molecular Dynamics simulation of riboswitch aptamer domain. (A)** Root-mean square deviations (RMSD) over 1000 ns, and (B) region-wise root-mean square fluctuations (RMSF) of the aptamer domain. Six different simulations were carried out. Apo crystal 6UBU with native ligand removed (black), crystal with native guanine (red), crystal with docked guanine (green), crystal with docked guanosine (blue), crystal with docked SK4 (purple), ensemble structure with docked SK4 (cyan).

interacting hydrogen bonds was found to be greater for guanine in comparison to guanosine and SK4. In the native crystal structure, guanine makes hydrogen bonds with residues U8, U33, U37, and C60 of the riboswitch. Simulation trajectories show that SK4 maintains hydrogen bonds with these four residues although bonding interactions with nucleobases U33 and U37 were maintained during the simulation with a lower residence time in comparison to strongly interacting bonds with U8 and C60 residues (Table 1). The interactions with residues U33 and U37 (J2-3 region) were missing in case of guanosine and SK4 systems. However, SK4 binds to adjacent residues A38 and C39 via moderately interacting H-bonds due to binding pocket adjustments. At the same time, SK4 and guanosine compounds maintained interactions with adjacent residues A9, A10 (J1-2 region). Therefore, guanosine and SK4 yield novel interactions due to the presence of the bulky ribose sugar ring compared to guanine. Free energy results can mostly be attributed to the bonding interactions observed over the arc of a simulation. It may be possible that simulation times longer than 1000 ns are required to explore a larger conformational space and observe a higher number of stable interactions. Future work in our laboratory will also be focused on investigating the free energy and binding constants of SK4 to the aptamer domain of the riboswitch with a direct biochemical technique such as isothermal titration calorimetry.

## Conclusions

In our work with nucleoside analog SK4 and the guanine riboswitch, we have shown that SK4 binds to the aptamer domain of the guanine riboswitch and lowers the expression of guanine riboswitch mRNA without compromising the viability of *B. subtilis* cells. There are evolutionary advantages of regulating specific genes of an organism without providing the impetus for the bacterium to evolve. Our work with SK4 demonstrates that specific genes in bacteria can be effectively controlled by ligand analogs, offering a different mechanism of action compared to the mechanisms of existing drugs. In the next phase of our research, we plan to explore whether SK4 binds to additional RNA molecules and other macromolecular targets. An interesting extension of the work would also be to determine whether SK4 is effective in gene regulation in antibiotic resistant bacteria.

## Materials and methods

**Synthesis:** Reactions were performed in oven-dried flasks or glass microwave vessels and those requiring an inert atmosphere were conducted under high purity argon. [1]H and [13]C NMR spectra were measured with a Varian MR-400

**Table 1. Hydrogen bonding data from molecular dynamics simulations.**

| Simula-tion | Initial Bonding Residues | Strong H-bonds | Moderate H-bonds | Weak H-bonds |
|---|---|---|---|---|
| Docked Guanine | A7, U8, U33, U37, A38, C60, U61 | A7, U8, U33, U37, A38, C60, U61 | C36 | U35, U59 |
| Docked Guanosine | A7, U8, A10, U59, C60, U61 | U8, A9, A10, U59, C60 | A7, A38 | A5, U6, C36, C39, G58, U61, A62 |
| Docked SK4 | U6, A7, U8, A9, A10, U37, U59, C60, U61 | A7, U8, A9, U59, C60, U61 | U6, A10, U33, A38, A62 | C4, A5, C36, U37 |

Initial bonding residues are ligand contacts with RNA residues prior to the simulation. Strong, moderate and weak H-bonds refer to ligand's contacts after the completed simulation. The strength of a bond is determined by the length of simulation time over which the H-bond contact is maintained with an RNA residue.

MHz Spectrometer and referenced to $CHCl_3$ at 7.26 ppm and 77.0 ppm, respectively, or DMSO at 2.50 ppm. Thin layer chromatography was performed on 60-mesh silica plates purchased from Sorbent Technologies (XHL, UV254, 250mm). Purification was achieved using CombiFlash® (Teledyne ISCO) Rf 150 Flash Chromatography System on RediSep® Gold Normal-phase silica-gel columns. All chemicals utilized in the reactions were purchased from Sigma Aldrich. Anhydrous solvents were dried over activated 3 Å molecular sieves and stored under inert atmosphere as described by Williams and Lawton [36]. Liquid chromatography/mass spectrometry (LCMS) analysis was collected using Varian 212-LC/ 500-MS equipped with a Phenomenex Kinetic 2.6u C18 150 X 2.10 mm column.

**Synthesis of 2',3',5'-tri-O-acetylguanosine (SK1):** Guanosine (15.0 g, 53.0 mmol) and N, N-(dimethylamino) pyridine (0.71 g, 5.73 mmol) were added to a round-bottom flask and flushed with argon for 10 minutes. Triethylamine (57.9 mL, 418 mmol) and anhydrous acetonitrile (203 mL) were then added and the suspension was stirred. The suspension was cooled to 0°C and acetic anhydride (16.3 mL, 166 mmol) was then added dropwise and stirred vigorously until the mixture became homogeneous. The solution was then stirred for an additional 3 h at room temperature. The reaction was slowly quenched with methanol (17.3 mL) and the volume was reduced by approximately 2/3 under reduced pressure. Diethyl ether was added dropwise using dropping funnel set-up to induce the precipitation of a fine white powder. This product was then collected by vacuum filtration with a fine glass frit, washed with diethyl ether. The solid was stirred in acetone (225 mL) for an additional 2 h at 50 °C. The resulting solid was filtered hot and collected to yield 19.7 g (91% yield) of a fine white powder. [1]H NMR (400 MHz, DMSO-$d_6$) δ 10.70 (s, 1H, N6-H), 7.91 (s, 1H, C8-H), 6.54 (bs, 2H, $NH_2$), 5.98 (d, $J$ = 6.1, 1H), 5.78 (t, $J$ = 6.1,1H), 5.49 (dd, $J$ = 6.1, 4.3 Hz, 1H), 4.38–4.24 (m, 3H), 2.11 (s, 3H, OAc) 2.04 (s, 3H, OAc), 2.03 (s, 3H, OAc).

**Synthesis of 2′,3′,5′-tri-O-acetyl-6-chloroguanosine (SK2):** [34] SK1 (5.00 g, 12.2 mmol) and tetraethylammonium chloride (4.05 g, 24.4 mmol) were added to a round bottom flask and flushed with argon for 10 minutes. Anhydrous 6acetonitrile (34.4 mL) was added followed by a slow addition of $N$-$N$-dimethylaniline (1.55 mL 12.2 mmol) and $POCl_3$ (6.83 mL, 73.3 mmol). The flask was heated to 70 °C in an oil bath for one hour. The volume was then reduced by 2/3 under reduced pressure and inverse quenched by slowly adding the reaction solution to a beaker containing about 700 mL $H_2O$. It was vigorously stirred for 10 minutes. The organic layer was extracted with DCM (100 mL x 7), collected and then washed with $H_2O$ (~200 mL x 10) until the pH of the aqueous layer was approximately that of the water itself. The collected organic mixture was then dried over $MgSO_4$, filtered, and the solvent was removed under reduced pressure. The resulting crude material was purified using silica gel flash column chromatography (20:80 EtOAc:hexanes to 100% EtOAc, gradient), affording a 1.67 g (32% yield) of a yellow-orange powder. [1]H NMR (400 MHz, $CDCl_3$) δ 7.87 (s, 1H), 6.01 (d, $J$ = 5.0 Hz, 1H), 5.95 (t, $J$ = 5.0 Hz, 1H), 5.75 (t, $J$ = 5.0 Hz, 1H), 5.22 (s, 2H), 4.47–4.42 (m, 2H), 4.40–4.34 (m, 1H), 2.14 (s, 3H, OAc) 2.10 (s, 3H, OAc), 2.08 (s, 3H, OAc).

**Synthesis of 2-acetamido-6-chloro-9-(2′,3′,5′-tri-*O*-acetyl-*β*-D-ribofuranosyl)purine (SK3)** DMAP (2.38g, 19.50 mmol) and SK2 (1.67g, 3.90 mmol) were added to a round bottom flask equipped with a stir bar and flushed with argon for 10 minutes. Anhydrous $CH_2Cl_2$ (42mL) was added followed by a slow addition of acetyl chloride (19.50 mmol, 1.39mL) and pyridine (19.50 mmol, 1.57mL). The reaction was stirred overnight and followed by an inverse quench, by adding the reaction mixture to $H_2O$ (~500mL). The mixture was stirred vigorously for 10 minutes. The product was extracted with $CH_2Cl_2$ (100mL x 7), washed with $H_2O$ (200mL x 8), and dried over $MgSO_4$. Resulting crude material was purified by silica gel flash chromatography (50% EtOAc:hexanes to 100% EtOAc, gradient) to afford an orange-yellow powder (1.83g, 99%). Spectral data is consistent with the literature [37]. $^1H$ NMR (400 MHz, $CDCl_3$) δ 8.15 (bs, 1H), 8.12 (s, 1H, H8), 6.09 (d, *J* = 4.5 Hz, 1H, H1′), 5.89 (dd, *J* = 4.5, 5.6 Hz, 1H, H2′), 5.75 (t, *J* = 5.6 Hz, 1H, H3′), 4.52–4.39 (m, 3H, H4′, H5′), 2.46 (s, 3H, NAc), 2.15 (s, 3H, OAc) 2.10 (s, 3H, OAc), 2.09 (s, 3H, OAc). IR (thin film): 3439 (m, amide, NH), 3205 (m, C=C-H), 1728 (s, ester C=O), 1634 (s, amide C=O) cm$^{-1}$. LRMS (pos. ESI) *m/z*: 492.2 (M + Na), 470.1 (M + H).

**Synthesis of 2-acetamido-6-hydroxyamino-9-(2′,3′,5′-tri-o-acetyl-β-d-ribofuranosyl)purine (SK4):** Potassium hydroxide (511mg, 9.12 mmol) was crushed into fine particles with a glass rod and dissolved in EtOH (7.00mL). In a separate round bottom flask, EtOH (9.00mL) was heated to reflux and hydroxylamine hydrochloride (540mg, 7.77 mmol) was added and stirred until dissolved. The KOH solution was poured into the reaction flask, stirred vigorously for 3–5 minutes and gravity filtered into a dry, round bottom flask. SK3 was then added to the reaction flask (150mg, 0.319 mmol). The reaction was kept under reflux for one hour and slowly cooled to RT for one hour. The reaction was then cooled in a salt/ice bath to crystallize the product. The supernatant was then carefully decanted, the product washed with cold EtOH, and then dried under high vacuum to yield a light yellow/brown solid (65.0mg, 60%). $^1H$ NMR (400 MHz, DMSO-$d_6$) δ 10.36 (bs, 1H), δ 7.99 (s, 1H, H8), 7.76 (s, 1H, NH), 6.37 (s, 1H, NH), 5.70 (s, 1H, H1′), 4.40 (m, 1H, H2′), 4.11 (dd, *J* = 4.4 Hz, 1H, H3′), 3.86 (m 1H, H4′) 3.60 (dd, *J* = 12.1, 4.4 Hz), 1H, H5′), 3.51 (dd, *J* = 12.1, 4.0 Hz, 1H, H5′), 2.14 (s, 3H, NAc). $^{13}C$ NMR (101 MHz, DMSO-$d_6$) δ 170.2, 166.2, 135.7, 115.7, 111.9, 86.9, 81.4, 73.2, 70.3, 64.0, 23.9. IR (thin film): 3600–2500 (br, s, ribose OH), 1693 (s, amide C=O) cm$^{-1}$. LRMS (neg. ESI) *m/z*: 375.0 (M + Cl), 340.1 (M$^+$), 339.1 (M - H), 297.0 (M - Ac).

**Beta-Galactosidase Assay:** Stock solutions of Guanine (Sigma Aldrich), Guanosine (Sigma Aldrich), and SK4 were prepared in DMSO. *Bacillus subtilis* cells with guanine riboswitch gene fused upstream of promotorless *lacZ* reporter gene were used to quantify riboswitch mRNA expression *in vivo*. *B. subtilis* cells were provided courtesy of Dr. Ronald Breaker (Yale University). Chloramphenicol (5 mg/mL) is the selection antibiotic. Cells were grown overnight in a chemically defined minimal medium (CDM) with constant shaking (220 rpm) at 37ºC. Cells at an OD600 ~ 0.1 were incubated with varying concentrations guanine, guanosine, and SK4. Cells were grown with constant shaking for 8 hours at 37ºC in a 96-well microplate. Beta galactosidase expression was measured using the standard Miller Unit assay [35]. Samples were run in five independent replicates. To test for statistical differences in expression between the different treatments, we used an analysis of variances implemented as a linear model in R v4.4.1. We used the Miller Unit concentration as our response variable and treatments as explanatory variables. We tested for statistical differences between treatments using the post hoc Tukey Test and bar plots were drawn in R's ggplot2.

**Kirby Bauer Disk Diffusion Assay:** Antibiotic susceptibility experiments were carried out using the Kirby Bauer disk diffusion assay [38]. Briefly, *B. subtilis* cells at an OD = 0.1 were spread using glass beads on LB agar plates with chloramphenicol. Subsequently, 9 mm filter disks impregnated with 25 µl of ligands at different concentrations were placed in the agar plates. The bacteria were grown overnight in a 37ºC incubator and images of the plates were taken.

**Quantitative PCR of mRNA transcripts:** *B. subtilis* cells were grown in CDM with various ligands (SK4, guanine, guanosine) for 8 hours with constant shaking (220 rpm) at 37ºC. Bacterial cells were pelleted and total RNA was extracted using the Direct-zol RNA miniprep kit from Zymo Research. After extraction, RNA samples were purified using phenol:chloroform:isoamyl extraction and ethanol precipitation. RNA samples were treated with DNAse and amplified using A probe-based assay. Briefly, cDNA synthesis was carried out followed by DNA amplification using the PrimeTime One-Step

RT-qPCR MasterMix (IDT DNA Inc.). *GAPDH* gene in *B. subtilis* was used as an internal control. All samples were run in quadruplicates and each reaction included GAPDH as an internal control to assess variability in our methodology. Primers and probes were also purchased from IDT DNA. Sequences are listed in the 5' to 3' direction and are shown below:

*xpt* forward primer: CAG AAA GCC AAA TCG CAG TG

*xpt* reverse primer: ATT CCC GCA ATA GAA GCT CC

*xpt* Probe:/56-FAM/ CACAATCGA/ZEN/ CACAAGCCCGTGC/3IABkFQ/

*GAPDH* forward primer: GCT TGC CCT GTC CAG TTA AT

*GAPDH* reverse primer: GCT CAG CTG CAC CCT TTA G

*GAPDH* Probe:/5SUN/ CCTGCCCTT/ZEN/ TGAGTTTGATGATGCTG/3IABkFQ/

To test for statistical differences in gene expression between the different treatments, we used an analysis of variances implemented as a linear model in R v4.4.1. We used Cq score as a proxy for gene expression as our response variable and treatments as explanatory variables. We also confirmed that there were no statistical differences between the internal controls, which were removed from subsequent analysis. We tested for statistical differences between treatments using the post hoc Tukey Test and bar plots were drawn in R's ggplot2. Finally, we estimated the two-fold changes in gene expression using the ΔΔCq method.

**Molecular Docking:** The crystal structure (PDB ID 6UBU) of guanine bound G-riboswitch (guanine riboswitch) solved at 1.6 Å was considered as the starting structure for the control apo system [26]. This structure is considered as a receptor after removing the prebound guanine ligand and three molecules (Guanine, Guanosine and SK4) are considered as ligands in the docking process using the DOCK6 program. Hydrogen atoms were eliminated, and UCSF Chimera was utilized to prepare the guanine riboswitch receptor. The DMS program in the DOCK6 software package was utilized to calculate the solvent-accessible surface of the riboswitch binding site with a probe radius of 1.4 Å. Receptor spheres were generated through the SPHGEN program, with selection criteria limited to spheres within 10 Å from the positions of the prebound guanine coordinates. A grid box enclosing the selected spheres was created, with an additional 5 Å added in each dimension. Ligand flexibility was considered in the docking process using the DOCK6 module, and the results were presented as grid scores. **Ensemble Docking:** Trajectories of different systems mentioned above were subjected to RMSD based clustering. The RMSD cut-off of 1 Å was used for the clustering using the dbscan method of cpptraj module of AmberTools 17. A total of 9 representative conformations for each ensemble were obtained through RMSD-based clustering and they were considered for the ensemble docking approach.

**Molecular Dynamics Simulations:** MD simulations of six systems namely 6UBU-NoLig, 6UBU-Crystal, 6UBU-GUA-docked, 6UBU-GNS-docked, 6UBU-SK4-docked and 6UBU-Ensemble-SK4-docked were carried out for 1 µs each. The simulations were performed using the AMBER20 simulation package [39]. The force field parameters were generated using the AMBER OL3 force field for the riboswitch RNA and gaff2 force field for the ligands [40,41]. All the systems were neutralized by addition of Na+ ions and were solvated using the TIP3P water model. The minimization was performed for 30000 steps in two successions. Initially, steepest descent minimization was performed for 20000 steps, the remaining steps were performed using the conjugate gradient method. In the next step the temperature was gradually increased to 300 K for the solvent, where in the solute was held using force restraints. Once the solvent attained the desired temperature, the entire simulation system was heated to achieve 300 K. The temperature was maintained at 300 K using the Langevin thermostat. The hydrogen restraints were taken care of by the SHAKE algorithm. The system was equilibrated under the NPT ensemble for 1 ns by maintaining the temperature and pressure at 300 K and 1 atm, respectively. A production run of 1 µs for each of the systems was performed.

**Analysis:** Root mean square deviation (RMSD) calculations were performed using the CPPTRAJ module of Amber-Tools17. The calculation of root mean square fluctuations (RMSF) of riboswitch was done for just the heavy atoms. The average structures for generating snapshots for the MD simulations were also obtained using CPPTRAJ. The donor-acceptor heavy atoms lying within 3.3 Å and at an angle of ≥135° were considered as hydrogen bonds. The distance

calculation between the two heavy atoms was performed using the CPPTRAJ module. VMD and UCSF Chimera were used for visualizing the MD trajectories and generating images. The MMPBSA.py module of AmberTools17 was used for calculating the free energy of binding between the ligands and the guanine riboswitch. Getcontacts was used for calculating the ligand interactions with the riboswitch.

## Supporting information

**S1 Fig. LCMS (ESI positive ion) spectrum for SK3.**
(TIF)

**S2 Fig. LCMS (ESI negative ion) spectrum of SK4.**
(TIF)

**S3 Fig.** $^1$H NMR spectrum of SK4 in DMSO-d6.
(TIF)

**S4 Fig.** $^{13}$C NMR spectrum of SK4 in DMSO-d6.
(TIF)

**S5 Fig. Infrared spectrum of SK4 (thin film).**
(TIF)

**S6 Fig. Antibiotic susceptibility studies were done using Kirby Bauer disk diffusion assay.** This figure shows that *B. subtilis* cells are viable at 500 micromolar guanine, guanosine, and SK4. Neomycin (with a zone of inhibition) is shown as a positive control.
(TIF)

**S7 Fig. Antibiotic susceptibility studies were done using Kirby Bauer disk diffusion assay.** This figure shows that *B. subtilis* cells are viable (no zones of inhibition are observed) upto a concentration of 500 micromolar SK4.
(TIF)

## Acknowledgments

The authors would like to thank the faculty members and students of the Bard College Chemistry and Biochemistry Program for their valuable contributions. We also appreciate the constructive feedback from the reviewers and editors, which has helped improve this manuscript.

## Author contributions

**Conceptualization:** Swapan Jain.

**Data curation:** Swapan S. Jain.

**Formal analysis:** Swapan S. Jain, Emily C. McLaughlin, Gabriel G. Perron, Mallikarjunachari Uppuladinne, Silvie H. Lundgren.

**Funding acquisition:** Swapan S. Jain.

**Investigation:** Swapan S.Jain, Emily C. McLaughlin, Seoyoung Kim, Katherina Gindinova, Silvie H. Lundgren, Liad Elmelech.

**Methodology:** Swapan S. Jain, Emily C. McLaughlin, Mallikarjunachari Uppuladinne.

**Project administration:** Swapan S. Jain, Emily C. McLaughlin.

**Resources:** Swapan S. Jain, Uddhavesh Sonavane, Rajendra Joshi.

**Software:** Swapan S. Jain, Gabriel G. Perron, Mallikarjunachari Uppuladinne, Uddhavesh Sonavane.

**Supervision:** Swapan S. Jain, Emily C. McLaughlin, Korrapati Narasimhulu.

**Validation:** Swapan S. Jain, Emily C. McLaughlin, Gabriel G. Perron, Mallikarjunachari Uppuladinne, Uddhavesh Sonavane.

**Visualization:** Swapan S. Jain, Mallikarjunachari Uppuladinne.

**Writing – original draft:** Swapan S. Jain, Emily C. McLaughlin, Mallikarjunachari Uppuladinne.

**Writing – review & editing:** Swapan S. Jain.

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
