## [Decision Letter · Decision Letter 0]

5 Jan 2025

PONE-D-24-53210Inhibition of xpt Guanine Riboswitch by a Synthetic Nucleoside AnalogPLOS ONE

Dear Dr. Jain,

Thank you for submitting your manuscript to PLOS ONE. After careful consideration, we feel that it has merit but does not fully meet PLOS ONE’s publication criteria as it currently stands. Therefore, we invite you to submit a revised version of the manuscript that addresses the points raised during the review process.

We look forward to receiving your revised manuscript.

Kind regards,

Shailza Singh, Ph.D

Academic Editor

PLOS ONE

Journal Requirements:

“The authors acknowledge financial support from Research Corporation for Science Advancement Award # 21054 to SSJ.  Generous funding for this project is also provided by the Office of Undergraduate Research at Bard College, Bard Summer Research Institute, and the Chemistry & Biochemistry Program at Bard College.”

“The authors acknowledge financial support from Research Corporation for Science Advancement Award # 21054 to SSJ.  Generous funding for this project is also provided by the Office of Undergraduate Research at Bard College, Bard Summer Research Institute, and the Chemistry & Biochemistry Program at Bard College. “ 

“The authors acknowledge financial support from Research Corporation for Science Advancement Award # 21054 to SSJ.  Generous funding for this project is also provided by the Office of Undergraduate Research at Bard College, Bard Summer Research Institute, and the Chemistry & Biochemistry Program at Bard College.”

Reviewers' comments:

Reviewer's Responses to Questions

**Comments to the Author**

1. Is the manuscript technically sound, and do the data support the conclusions?

Reviewer #1: No

Reviewer #2: Yes

2. Has the statistical analysis been performed appropriately and rigorously? 

Reviewer #1: No

Reviewer #2: Yes

3. Have the authors made all data underlying the findings in their manuscript fully available?

Reviewer #1: Yes

Reviewer #2: Yes

4. Is the manuscript presented in an intelligible fashion and written in standard English?

Reviewer #1: Yes

Reviewer #2: Yes

5. Review Comments to the Author

Reviewer #1: Major comments:

1. In Figure 4, are these results of three or four independent experiments? Statistical analysis to be done again. If you compare, guanine, guanosine and SK4 molecule, drastic change is not seen. The values of SK4 are close to values of Guanine and guanosine

2. The viability of Bacillus subtilis cells is not impacted by SK4. How these analogs can function as alternatives to commercially available antibiotics is not understood.

3. The messenger RNA of the xpt gene in bacteria codes for xanthine phosphoribosyltransferase enzyme which is important in the metabolic pathway of bacteria catalyzing the synthesis of xanthine monophosphate from xanthine. If that is so the viability of bacteria should have been affected.

4. Which specific genes does SK4 regulate? The authors have not checked time dependent regulation of expression of the riboswitch

5. Primer sequences on pg 15 are without direction. 5’ to 3’ direction needs to be specified.

Authors have not explained how these analogs will tackle drug resistance cases.

Reviewer #2: Reviewer comments:

The manuscript entitled “Inhibition of xpt Guanine Riboswitch by a Synthetic Nucleoside Analog” by Jain et al., is aimed at analyzing SK4, a novel nucleoside analog using a two-pronged in vitro and in silico approaches that is capable of inhibiting the transcription of riboswitch mRNA. The experiments followed were appropriate. However, the following points needs to be addressed for better understanding of the readers and to improve the quality of the manuscript.

Major comments:

Introduction:

1. What potential pharmaceutical applications does SK4 possess? Discuss in detail.

2. What is the significance of synthesizing SK4, a nucleoside analog?

Results and discussion:

3. How does SK4 binding to guanine riboswitches be utilized in therapeutic contexts?

4. What statistical analysis method was used to determine the significance of the results for in vitro experiments?

5. The results and discussion can be given as two separate sub-headings.

6. The results provided for qPCR can be interpreted as xpt transcripts normalized with GAPDH in the axis title of the graph.

Conclusion:

7. Explain the further research or testing is needed to explore the potential applications of SK4.

Minor comments:

1. In page 15, line no:359, the term 25 microliters can be changed as 25 μl.

2. The authors are suggested to check for the language correction throughout the manuscript.

3. The authors are advised to format the references uniformly as per the “author guidelines” of the journal.

6. PLOS authors have the option to publish the peer review history of their article (what does this mean? ). If published, this will include your full peer review and any attached files.

**Do you want your identity to be public for this peer review?** For information about this choice, including consent withdrawal, please see our Privacy Policy .

Reviewer #1: No

Reviewer #2: **Yes: ** Sridhar Muthusami

---

## [Author Response · Author response to Decision Letter 0]

17 Feb 2025

We sincerely appreciate the thoughtful and very helpful comments by the reviewers. Please see below each of the reviewer comments and our response directly underneath each comment. These changed have been revised in the manuscript as indicated below.

Reviewer #1 Major Comments

1. In Figure 4, are these results of three or four independent experiments? Statistical analysis to be done again. If you compare, guanine, guanosine and SK4 molecule, drastic change is not seen. The values of SK4 are close to values of Guanine and guanosine

Response: It is already stated in the caption for Figure 4 that quadruplicates were tested. We have provided further clarifications about the statistical analysis in Figure 4 caption and in the Materials and Methods section. Regarding the results, we do not claim that there is a drastic change. We only state that there are statistically significant changes, supported by our analysis of quadruplets testing the differences between every treatment. We note that a single quantitation cycle change in qPCR is an important change. We have shown the quantitation cycle values in the description of results for Figure 4. We have also now included a fold-change in mRNA expression between SK4 and the other two compounds (guanine and guanosine). The fold change determination was carried out using the ΔΔCq method as explained in the methods section.

2. The viability of Bacillus subtilis cells is not impacted by SK4. How these analogs can function as alternatives to commercially available antibiotics is not understood.

Response: We have edited the manuscript and removed language to avoid any inference that SK4 will function as an alternative to commercially available drugs or as a means to target antibiotic resistant bacteria. Our goal in this work is to show that modulation of riboswitch expression by SK4 provides an alternative mechanism of gene regulation at the mRNA level. Please also see response to the next item (point # 3).

3. The messenger RNA of the xpt gene in bacteria codes for xanthine phosphoribosyltransferase enzyme which is important in the metabolic pathway of bacteria catalyzing the synthesis of xanthine monophosphate from xanthine. If that is so the viability of bacteria should have been affected.

Response: There are other mechanisms in bacteria such as the purine salvage pathway (adenine and guanine phosphoribosyl transferase genes) which can compensate for the loss of xanthine phosphoribosyl transferase function. Bacteria can also take up purines from their environment and survive. This is the reason why the viability of the bacteria is not impacted. Nevertheless, we agree that this is a good point and we have changed the language in the Introduction section to remove the words “important in the metabolic pathway”

4. Which specific genes does SK4 regulate? The authors have not checked time dependent regulation of expression of the riboswitch

Response: SK4 regulates the mRNA of the xanthine phosphoribosyl transferase (xpt) gene in bacteria. This is now clearly indicated in the introduction and in the abstract section. In this work, we have synthesized a novel compound and tested its binding and gene regulation via biochemical and computational studies. We have designed SK4 to specifically bind the aptamer region of xpt mRNA. It is a widely accepted practice in the field that riboswitch expression studies are carried out whereby incubation with compounds starts when the OD600 bacteria is 0.1. Cells are subsequently grown for a defined period of time (8 hours in this work). At that time, cells are analyzed to determine the expression of specific genes. We have suggested additional work in the conclusions section of the manuscript but we feel that the time-dependent studies are not necessary to reach our conclusions and they are out of the scope of this work.

5. Primer sequences on pg 15 are without direction. 5’ to 3’ direction needs to be specified.

Authors have not explained how these analogs will tackle drug resistance cases.

Response: The direction of primer sequences is already indicated in the methods section in the paragraph above the sequences. We have changed the manuscript language in the abstract, introduction, and conclusions section to avoid any confusion that these analogs will tackle drug resistant bacteria. Our goal in this work is to validate riboswitches as drug targets and to show that alternative mechanisms of gene regulation in bacteria are plausible.

Reviewer #2 Major Comments

Introduction:

1. What potential pharmaceutical applications does SK4 possess? Discuss in detail.

Response: SK4 offers a unique regulation mechanism in bacteria (targeting of mRNA) which differs from existing bactericidal therapeutic drugs. Nearly all of the existing drugs target bacteria by binding to proteins or enzymes (DNA replication, cell wall synthesis, protein synthesis). Targeting of riboswitches by analogs is regulation at the mRNA level. We have changed the introduction to indicate that the focus of SK4 is to explore an alternate mechanism of regulation in bacteria and to validate riboswitches as drug targets.

2. What is the significance of synthesizing SK4, a nucleoside analog?

Response: The significance of synthesizing SK4 is to create a molecule that could specifically bind to the riboswitch mRNA and lower its expression. SK4 design and synthesis was motivated by our previous work in 2015 and work done by Breaker and other groups referenced in the manuscript. We have changed the introduction section to show that SK4 was synthesized in a manner that allows for additional bonding interactions with the mRNA. We have also moved Figure 1 (riboswitch and SK4 structure) below the paragraph in the introduction section which describes the significance of SK4 synthesis.

Results and discussion:

3. How does SK4 binding to guanine riboswitches be utilized in therapeutic contexts?

Response: Exploration of SK4 as a therapeutic molecule is out of the scope of this work. We have carried out organic synthesis of SK4 and explored its binding to the riboswitch RNA using in vitro, in vivo, and computational studies. Future work may look at cellular studies with antibiotic resistant bacteria and we have mentioned this in the conclusions section.

4. What statistical analysis method was used to determine the significance of the results for in vitro experiments?

Response: We have edited captions for Figure 3 and Figure 4 to indicate that significance differences were estimated using an analysis of variance followed by a post-hoc Tukey test. We have also added the following language in the Materials and Methods section to further define the statistical methods for the in vitro qPCR experiments.

“To test for statistical differences in gene expression between the different treatments, we used an analysis of variances implemented as a linear model in R v4.4.1. We used Cq score as a proxy for gene expression as our response variable and treatments as explanatory variables. We also confirmed that there were no statistical differences between the internal controls, which were removed from subsequent analysis. We tested for statistical differences between treatments using the post hoc Tukey Test and bar plots were drawn in R’s ggplot2. Finally, we estimated the two-fold changes in gene expression using the ΔΔCq method.”

5. The results and discussion can be given as two separate sub-headings.

Response: Due to the interdisciplinary nature of the work, we found it helpful to contextualize our results by writing the results and discussion under a single heading. We have found other instances in PLOS ONE where authors have chosen to write the results and discussion under a single heading and we feel that this choice also fits our manuscript.

6. The results provided for qPCR can be interpreted as xpt transcripts normalized with GAPDH in the axis title of the graph.

Response: This is a good point. We have edited the Materials and Methods section for the qPCR experiments to further define the role of GAPDH as an internal control. We have also provided the quantitation cycle values for GAPDH samples as well as guanosine, guanine, and SK4 within the manuscript. After comparing the GAPDH results for all reactions, we found no statistical differences between them. Therefore, we do not feel that there is a need to normalize the values in the axis title.

Conclusion:

7. Explain the further research or testing is needed to explore the potential applications of SK4.

Response: In the conclusion section, we have discussed future work which will explore SK4 binding to other RNA molecules and macromolecular targets. Studies with antibiotic resistant bacteria and SK4 are also an additional line of inquiry that has been mentioned in the conclusions section.

Minor comments:

1. In page 15, line no:359, the term 25 microliters can be changed as 25 μl.

Response: This has been changed as suggested.

2. The authors are suggested to check for the language correction throughout the manuscript.

Response: Manuscript was edited for clarity and flow. Modifications are indicated in the track changes mode.

3. The authors are advised to format the references uniformly as per the “author guidelines” of the journal.

Response: The references have been formatted uniformly with journal abbreviations as referenced in the NCBI databases and references are cited in Vancouver style.

---

## [Decision Letter · Decision Letter 1]

19 Mar 2025

Inhibition of xpt Guanine Riboswitch by a Synthetic Nucleoside Analog

PONE-D-24-53210R1

Dear Dr. Jain,

We’re pleased to inform you that your manuscript has been judged scientifically suitable for publication and will be formally accepted for publication once it meets all outstanding technical requirements.

Kind regards,

Shailza Singh, Ph.D

Academic Editor

PLOS ONE

Additional Editor Comments (optional):

Reviewers' comments:

Reviewer's Responses to Questions

**Comments to the Author**

1. If the authors have adequately addressed your comments raised in a previous round of review and you feel that this manuscript is now acceptable for publication, you may indicate that here to bypass the “Comments to the Author” section, enter your conflict of interest statement in the “Confidential to Editor” section, and submit your "Accept" recommendation.

Reviewer #1: All comments have been addressed

Reviewer #2: All comments have been addressed

2. Is the manuscript technically sound, and do the data support the conclusions?

Reviewer #1: Yes

Reviewer #2: Yes

3. Has the statistical analysis been performed appropriately and rigorously? 

Reviewer #1: Yes

Reviewer #2: Yes

4. Have the authors made all data underlying the findings in their manuscript fully available?

Reviewer #1: Yes

Reviewer #2: Yes

5. Is the manuscript presented in an intelligible fashion and written in standard English?

Reviewer #1: Yes

Reviewer #2: Yes

6. Review Comments to the Author

Reviewer #1: All the queries have been addressed point to point. The manuscript can be accepted in the current form

Reviewer #2: Authors addressed all the previous concerns raised. Now the manuscript is suitable for consideration for publication in PLoS

7. PLOS authors have the option to publish the peer review history of their article (what does this mean? ). If published, this will include your full peer review and any attached files.

**Do you want your identity to be public for this peer review?** For information about this choice, including consent withdrawal, please see our Privacy Policy .

Reviewer #1: **Yes: ** Sushma Singh

Reviewer #2: **Yes: ** Sridhar Muthusami

---

## [Editor Report · Acceptance letter]

PONE-D-24-53210R1

PLOS ONE

Dear Dr. Jain,

I'm pleased to inform you that your manuscript has been deemed suitable for publication in PLOS ONE. Congratulations! Your manuscript is now being handed over to our production team.

Kind regards,

on behalf of

Dr. Shailza Singh

Academic Editor

PLOS ONE